# Ex Vivo Optimization of Donor Lungs with Inhaled Sevoflurane during Normothermic Ex Vivo Lung Perfusion (VITALISE): A Pilot and Feasibility Study in Sheep

**DOI:** 10.3390/ijms25042413

**Published:** 2024-02-19

**Authors:** Timo Steinkühler, Shuqi Yang, Michiel A. Hu, Jayant S. Jainandunsing, Neeltina M. Jager, Michiel E. Erasmus, Michel M. R. F. Struys, Dirk J. Bosch, Matijs van Meurs, Matthieu Jabaudon, Damien Richard, Wim Timens, Henri G. D. Leuvenink, Gertrude J. Nieuwenhuijs-Moeke

**Affiliations:** 1Department of Anesthesiology, University Medical Center Groningen, 9713 GZ Groningen, The Netherlands; 2Department of Thoracic Surgery, University Medical Center Groningen, 9713 GZ Groningen, The Netherlands; 3Department of Basic and Applied Medical Sciences, Ghent University, 9000 Ghent, Belgium; 4Department of Critical Care, University Medical Center Groningen, 9713 GZ Groningen, The Netherlands; 5Department of Perioperative Medicine, University Hospital Clermont-Ferrand, 63001 Clermont-Ferrand, France; 6Institute of Genetics, Reproduction & Development, University Clermont Auvergne, 63001 Clermont-Ferrand, France; 7National Institute of Health and Medical Research (INSERM), National Center for Scientific Research (CNRS), 75794 Paris, France; 8Department of Pharmacology and Toxicology, University Hospital Clermont-Ferrand, University Clermont Auvergne, 63001 Clermont-Ferrand, France; 9Department of Pathology and Medical Biology, University Medical Center Groningen, 9713 GZ Groningen, The Netherlands; 10Department of Surgery, University Medical Center Groningen, 9713 GZ Groningen, The Netherlands

**Keywords:** ex vivo lung perfusion, lung transplantation, ischemia reperfusion injury, anesthesia, sevoflurane, volatile anesthetics

## Abstract

Volatile anesthetics have been shown in different studies to reduce ischemia reperfusion injury (IRI). Ex vivo lung perfusion (EVLP) facilitates graft evaluation, extends preservation time and potentially enables injury repair and improvement of lung quality. We hypothesized that ventilating lungs with sevoflurane during EVLP would reduce lung injury and improve lung function. We performed a pilot study to test this hypothesis in a slaughterhouse sheep DCD model. Lungs were harvested, flushed and stored on ice for 3 h, after which EVLP was performed for 4 h. Lungs were ventilated with either an FiO_2_ of 0.4 (EVLP, n = 5) or FiO_2_ of 0.4 plus sevoflurane at a 2% end-tidal concentration (C_et_) (S-EVLP, n = 5). Perfusate, tissue samples and functional measurements were collected and analyzed. A steady state of the target C_et_ sevoflurane was reached with measurable concentrations in perfusate. Lungs in the S-EVLP group showed significantly better dynamic lung compliance than those in the EVLP group (*p* = 0.003). Oxygenation capacity was not different in treated lungs for delta partial oxygen pressure (PO_2_; +3.8 (−4.9/11.1) vs. −11.7 (−12.0/−3.2) kPa, *p* = 0.151), but there was a trend of a better PO_2_/FiO_2_ ratio (*p* = 0.054). Perfusate ASAT levels in S-EVLP were significantly reduced compared to the control group (198.1 ± 93.66 vs. 223.9 ± 105.7 IU/L, *p* = 0.02). We conclude that ventilating lungs with sevoflurane during EVLP is feasible and could be useful to improve graft function.

## 1. Introduction

The development of ex vivo lung perfusion (EVLP) has contributed to a two-fold increase in lung transplantations performed worldwide during the last decade [1]. EVLP allows for evaluation of the quality of potentially transplantable lungs and has led to an expansion of the donor pool and extension of the preservation time over 12 h [2,3,4,5]. In addition, EVLP provides a window of opportunity to treat injury, enhance repair and improve the quality of the lungs under semi-physiological conditions.

Ischemia reperfusion injury (IRI) is an important underlying mechanism of short- and long-term graft injury and currently inevitable in transplantation [6,7,8,9,10,11]. IRI is associated with a rapid increase in reactive oxygen species, leading to cellular and tissue injury, lipid membrane peroxidation, secretion of cytokines and release of damage-associated molecular patterns (DAMPs) [6,7]. These DAMPs subsequently activate compounds of the innate immune system, like neutrophils and macrophages as well as mediators of the complement system, further damaging the graft [8,9,10]. In addition, endothelial cell dysfunction, disintegration of the glycocalyx and disruption of the endothelial barrier occur. This leads to the exposure of integrins and endothelial surface receptors facilitating leukocyte infiltration and endovascular fluid leakage, resulting in interstitial and alveolar edema [7,11].

Volatile anesthetic agents, like sevoflurane, have been shown to reduce IRI in various organs [12,13,14,15,16], including the lungs [17,18,19,20,21], in animals and in humans. The use of sevoflurane may protect organs against IRI by preventing mitochondrial permeability transition pores from opening, preserving the glycocalyx, upregulating and stabilizing hypoxic inducible factors (HIFs) and directly affecting circulating immune cells [22,23,24,25,26,27,28,29].

Taken together, the benefits of sevoflurane in attenuating IRI combined with the therapeutic window of opportunity provided by EVLP may improve graft function and thereby improve the recipient outcome. We hypothesized that the ventilation of donor lungs with sevoflurane during EVLP would reduce injury and improve the quality of donor lungs. To test this, we designed a multistep project from bench to bedside. In this pilot project (step 1), we evaluated the feasibility of ventilating lungs with sevoflurane during EVLP in our pre-clinical slaughterhouse sheep model, resembling donation after circulatory death.

## 2. Results

We performed a total of 10 experiments in which 5 lungs were ventilated with sevoflurane (S-EVLP) and 5 without (EVLP). Lungs in both groups proved to be able to maintain stable EVLP parameters and lung function. Baseline characteristics of the lungs and procedure are listed in Table 1.

### 2.1. Sevoflurane Measurements

A steady state of the target C_et_ of 2% was reached after approximately ten minutes. This was maintained with a median administration of 2.8 mL/h (Appendix A). The measured sevoflurane levels (C_et_ and in perfusate samples) are depicted in Figure 1. Levels in perfusate markedly decreased at 120, 180 and 240 min. These decreases were reflected to a lesser extent in the C_et_. These timepoints correspond with the functional evaluation timepoints.

### 2.2. Functional Lung Evaluation

Functional lung measurement data are presented in Figure 2A,B and Table 2. The S-EVLP group showed a significantly higher dynamic lung compliance over time compared to the EVLP group (*p* = 0.003). There were no significant between-group differences in oxygenation capacity over time (Figure 2B), nor between the end and beginning (∆pO_2_, Table 2) or after 4 h of EVLP. No significant between-group difference was found in lung weight or fluid content between the beginning and end of EVLP.

### 2.3. Histology

Exemplary tissue samples, scored right before the start of EVLP (after SCS) and after EVLP, are shown in Figure 3. Lungs from both groups showed reactive changes but no substantial histological injury before EVLP, with normal alveolar septa, limited presence of leukocytes, limited alveolar collapse and no alveolar hemorrhage or edema. After EVLP, the histological images displayed a similar histology, with more pronounced expansion of the alveoli, likely caused by the ventilation by the EVLP procedure. Since most leukocytes were flushed out, we did not quantify any lung injury score.

### 2.4. Perfusate Analysis

Cytokine perfusate levels (Figure 4) at the start of EVLP were below detection levels. Levels of ASAT, LDH and TNF-α gradually increased over time in both groups. The increase in ASAT levels was significantly less pronounced in the S-EVLP group compared to the EVLP group (*p* = 0.02). No significant differences between the groups were found in levels of LDH and TNF-α. Levels of hyaluronan in the S-EVLP group increased over time, with a significant difference in levels at T180 (*p* = 0.008) compared to the EVLP group. This difference, however, was non-significant at T120 and T240.

### 2.5. Gene Expression of Cytokines in Tissue Samples

Gene expression of cytokines in lung tissue samples, measured before (after SCS) and after EVLP, is shown in Table 3 and Appendix A. A significant decrease in Ang1 was observed in the S-EVLP group in contrast to an increase in the EVLP group (*p* = 0.047). No significant differences between both groups were found in other measurements. The mRNA expression of endothelial integrity molecules VE cadherin and Tie2/TEK, endothelial adhesion molecules PECAM-1 and ICAM-1, endothelial vascular leakage-related molecules Ang2 and VEGF-α, hypoxia-related molecule HIF1a and neutrophil-related molecule MPO showed no significant difference after EVLP when compared to before. With respect to endothelial adhesion molecules VCAM and E-selectin, and proinflammatory cytokines IL-6 and TNF-α, the mRNA expression in both groups increased after EVLP.

## 3. Discussion

In this pilot study, we have shown that ventilation of the lungs with sevoflurane during EVLP is feasible. Sevoflurane C_et_ was stable over time while being readily taken up in the preservation solution.

Sevoflurane levels in the perfusate markedly decreased at 120, 180 and 240 min, which was reflected in the C_et_ values (Figure 1). At these timepoints, the oxygenation capacity of the lungs was evaluated, and the perfusion solution was actively deoxygenated, whilst the FiO_2_ was set to 100%. This might have caused a second gas effect, leading to an increased outward diffusion of sevoflurane along with the outward diffusion of O_2_ in the membrane (de)oxygenator [31].

Dynamic lung compliance, together with oxygenation capacity, as indicated by the venous PO_2_-to-FiO_2_ (P/F) ratio during and at the end of EVLP, are frequently used outcome measures in clinical EVLP studies and important criteria for transplantation [32,33]. Dynamic lung compliance was higher in the sevoflurane-treated lungs than in untreated lungs. A potential explanation is that endothelial dysfunction and, therefore, extravasation of fluid leading to tissue edema, an important hallmark of IRI, is less pronounced in the sevoflurane-treated lungs. There was, however, no difference between groups regarding edema formation, not in total weight difference before vs. after EVLP nor in fluid content of the tissue samples before vs. after EVLP. Another explanation could be that sevoflurane acts as bronchodilator, resulting in higher dynamic compliance values and thereby enabling higher tidal volumes to be achieved during pressure-controlled ventilation [34].

P/F ratios were comparable to those observed in human EVLP protocols, where a P/F ratio above 400 mmHg is usually considered one of the acceptance criteria [1]. Oxygenation capacity seems potentially superior in treated lungs, although not indicated by a difference in delta pO_2_, but supported by a trend toward a better P/F ratio. Our pilot and feasibility study, however, was underpowered to provide evidence for this.

The discrepancy between changes in total lung weight and fluid content in tissue samples might be explained by a larger intravascular compartment at the end of EVLP compared to the start of EVLP due to hypoxic pulmonary vasoconstriction due to ischemia prior to EVLP.

ASAT, LDH and TNF-α levels increased over time during EVLP. This could have been due to ongoing secretion and wash-out from injured cells due to initial injury upon warm ischemia and/or due to ongoing injury during the time course of EVLP. Our data imply a beneficial effect of sevoflurane as the ASAT levels in the S-EVLP group increased to a lesser extent. LDH and TNF-α levels, however, showed no significant between-group difference. Various studies have shown that sevoflurane attenuates pulmonary IRI and inflammation [17,18,19,20]. In a rat EVLP model, Wang et al. [21] showed that adding sevoflurane directly to the perfusion system upon the start of EVLP was associated with reduced levels of IRI injury markers like TNF-α and LDH. This difference might be explained by the different administration route (directly into the perfusate), different timing of administration (directly upon start EVLP), different warm ischemia time (60 min) or the difference in chosen species. In contrast to our study, dynamic lung compliance was comparable between groups.

Levels of hyaluronan increased significantly over time in the S-EVLP group but did not in the EVLP group. Hyaluronan is an important component of the extracellular matrix and one of the compounds of the glycocalyx, from which it can be released upon IRI or oxidative stress [35]. In an experimental set up with ex vivo perfusion of guinea pig hearts, treatment with sevoflurane showed a protective effect on glycocalyx integrity [24,25]. Hyaluronan fragments have different biological characteristics based on their length of chain and molecular weight. Low-molecular-weight (LMW) hyaluronan is proinflammatory while the high-molecular-weight (HMW) hyaluronan promotes immunotolerance, tissue remodeling and repair [36,37]. Previous studies [36,37,38] have shown that the dynamic change in hyaluronan in EVLP is a highly complex process that can affect lung function and is correlated with other molecules such as matrix metalloproteinase 2 and matrix metallopeptidase 9.

In our study, it was unclear which proportion of hyaluronan in the S-ELVP group consisted of LMW and which was of HMW. Furthermore, the ELISA used (the only one available for sheep) cannot differentiate between HMW and LMW hyaluronan. Therefore, we cannot state whether the increase seen in the S-EVLP group is either beneficial or detrimental. Larger studies focusing on other extracellular matrix endothelial glycocalyx products should be considered.

Endothelial adhesion molecules are known to be upregulated upon IRI, and sevoflurane has shown to reduce this injury-induced expression [24]. We observed no difference in the expression of adhesion molecules (ICAM-1, PECAM-1, E-selectin, VCAM-1) in the S-EVLP group compared to the EVLP group. A potential protective point of engagement of sevoflurane is the ability to upregulate HIF, leading to the transcription of genes involved in repair and survival [27]. In our study, the changes in expression of HIF1α and VEGF-a in both groups showed no significant differences between groups at a transcriptional level. This might be due to the fact that tissue samples were obtained directly at the end of EVLP, which might be too early for these genes and proteins to be upregulated.

In the S-EVLP group, Ang1 expression decreased over time, in contrast to an increase seen in the EVLP group. The angiopoietin signaling system, similar to the VEGF family, belongs to the angiogenic factors. Tie2/TEK, as the ligand of Ang1 and Ang2, jointly regulates the integrity of the endothelium [39,40].

The histological slides showed no significant lung injury before or after EVLP. This most likely occurred due to flushing out inflammatory cytokines and/or immune cells during the flush and perfusion. This is in line with Medeiros et al. [41], who observed a trend of a decreased apoptotic cell count in the lung tissue after EVLP. More importantly, it is likely that injury occurring during EVLP is not immediately evident on histological analysis but happens at the sub-microscopic level, leading to increased capillary leakage with edema and perhaps subtle attempts at tissue repair with resident cells producing extracellular matrix proteins like fibronectin and hyaluronic acid. Lastly, the tissue samples obtained before EVLP are from a different location of the lungs than the post-EVLP samples, which also makes comparison difficult. The fact that we observed no notable (deterioration of) lung injury, however, supports the interpretation that our EVLP model does not cause or worsen significant short-term histological injury to the lungs.

Since this first step was a feasibility study, we chose to perform 4 h of perfusion, with a sevoflurane C_et_ of 2%, although it has been shown that EVLP can be performed for up to 12 h [4]. Accordingly, future studies could consider studying longer perfusion times, potentially enhancing the beneficial effects of sevoflurane. In addition, higher concentrations of sevoflurane should be investigated to study dose-dependent effects. In step 2 of the VITALISE project, we are currently comparing 0%, 2%, 4% and 6% C_et_ of sevoflurane in a randomized setting. Since sevoflurane has shown superior effects over isoflurane in pulmonary inflammation models and liver transplant models, we choose to use sevoflurane in this project [29,42]. Since desflurane has shown the least organ protective effects in various experiments and its use in clinical practice is being phased out due to its significant global warming potential, we do not find it justified to study its effect during EVLP [43]. Isoflurane could be taken into account in future experiments and compared to sevoflurane, in which the focus should lie on differences in their protective potential and specific mechanisms of action.

Our study has limitations. Firstly, since this was a feasibility study, groups were non-randomized. Furthermore, for logistical reasons, we started with the S-EVLP groups, which were tested at the end of winter, a season known to negatively impact lung quality due to pulmonary infections [44]. We observed a notable inter-individual variability in organ quality, judged by clinical assessment at the slaughterhouse and reflected in our results. This implies that we started with the worst lungs (S-EVLP), as reflected in higher gene expression of various adhesion molecules and Ang 1 and Ang 2 before EVLP compared to the non-treated lungs (Appendix A). Seasonal differences, age of the animal and geographical location of the farm are among the factors that we suspect influence organ quality. This emphasizes the need for randomization in future studies aiming to investigate the effects of sevoflurane administration during EVLP, in order to account for the observed variability. Also, although we achieved average flow levels of over 90 percent of the calculated target values (Table 1), there was notable variability in the achieved flow levels, especially in the S-EVLP group. As mentioned, all S-EVLP experiments were performed first, which suggests that this was caused by a learning curve effect during the first experiments and that we were able to achieve more stable flow rates later during the study while performing the control experiments. This again emphasizes the need for randomization but also more elaborate training of the executing personnel for future experiments. Lastly, numbers per group were small and were not aimed at providing adequate statistical power to conclude that there is a clinical effect of sevoflurane on the measured indices of lung quality.

## 4. Materials and Methods

### 4.1. Animal Model and Treatment Groups

Our research group developed a model for testing sheep lungs obtained from a slaughterhouse (Kroon Vlees, Groningen, The Netherlands) [45], which means no additional animals are sacrificed for research. The setup and timeline are visualized in Figure 5. Sheep were terminated via a standardized procedure, followed by exsanguination, before the lungs were dissected. After retrieval, the lungs were inspected and ventilated with a resuscitation bag for 5 min to resolve atelectasis, whereafter the trachea was clamped to keep the lungs inflated. The heart and residual tissues were dissected, exposing the left atrium (LA) and pulmonary artery (PA). Subsequently, the lungs were flushed with 3 L (1 L anterograde, 2 L retrograde) cooled dextran-based electrolyte flush solution (Appendix A). Thereafter, the lungs were stored in flush-out solution and double-bagged in ice water. After transporting the lungs in Static Cold Storage (SCS) to our research lab, the LA (venous limb) and the PA (arterial limb) were cannulated while maintaining SCS.

The EVLP circuit (Figure 5), based on a Lung Assist Machine (XVIVO, Gothenburg, Sweden), was primed with 2 L of acellular-buffered dextran albumin solution perfusate (Appendix A). After 3 h of SCS, the lungs were placed on the Lung Assist and perfusion was started with 5–10% of the max flow (max flow = 20% of the calculated cardiac output, CO = 40% of the estimated bodyweight). Lungs were rewarmed by increasing the temperature of the perfusate gradually over time. Simultaneously, perfusion flow was gradually increased to 100% of the max flow at 60 min if the PA pressure remained <15–20 mmHg. 

When the perfusate outflow temperature reached 32 °C, the lungs were ventilated with 40% FiO_2_, either with (S-EVLP, n = 5) or without (EVLP, n = 5) the addition of sevoflurane at a 2% end-tidal concentration (C_et_). Sevoflurane was administered using the SedaConDa ACD-S device (Sedana Medical, Danderyd, Sweden) upon the start of ventilation, and an initial bolus of 0.5 mL was given to load the system. The ventilator (Servo-i: Maquet Critical Care, Solna, Sweden) was set to pressure-controlled ventilation with a positive end-expiratory pressure (PEEP) of 5 cmH_2_O, an inspiratory pressure of 15 cmH_2_O above PEEP, an inspiration-to-expiration (I:E) ratio of 1:2 and a respiratory rate of 7/min. PEEP was set to 10 cmH_2_O during a one-minute recruitment phase every 30 min. Deoxygenation gas (86% N_2_, 6% O_2_ and 8% CO_2_) was administered via the membrane oxygenator during the evaluation phases. The total duration of the EVLP was 4 h, after which the EVLP was stopped and samples and tissues were collected.

### 4.2. Inclusion/Exclusion Criteria

Inclusion criteria were warm ischemia time under 30 min, and lungs easily and evenly inflatable using a resuscitation bag with a PEEP valve set at 10 cmH_2_O. Exclusion criteria were cuts or prominent atelectasis, evident discoloration (e.g., purple indicating pneumonia), tracheal aspirations leaking and palpable nodules or other tissue irregularities.

### 4.3. Samples

Perfusate and tissue samples were obtained at standardized timepoints and from standardized locations (Appendix A [30] and Appendix A). Perfusate samples were collected, centrifuged (1800 rpm for 10 min at 4 °C) and stored at −80 °C until further analysis. For the sevoflurane measurements, perfusate samples were collected and centrifuged (1800 rpm for 10 min at 4 °C), and 1 mL was pipetted into a 10 mL headspace tube, which was immediately sealed with a Teflon cap and stored at −80 °C [30]. Tissue samples were divided into 5 parts. Samples for wet weight/dry weight (WW/DW) measurement were stored at 4 °C, weighed the next day and then dried in an Eppendorf oven (Avantor, VWR, Analog Heatblock, Amsterdam, The Netherlands) for 24 h at 100 °C before they were weighed again. A second part of the tissue was stored in 4% neutral buffered formalin, cleared in xylene and embedded in paraffin. Sections of 4 μm were cut and stained with hematoxylin and eosin. High-resolution images of the sections were taken with a C9600 NanoZoomer (Hamamatsu Photonics, Hamamatsu, Japan) and assessed by the Department of Pathology, University Medical Center Groningen. Tissue samples for mRNA analysis were stored at −80 °C.

### 4.4. Data and Sample Measurements

#### 4.4.1. Sevoflurane Measurements

Sevoflurane C_et_ was measured with an inline gas monitor (Datex-Ohmeda S/5, GE Healthcare, Chicago, IL, USA). Sevoflurane levels in the perfusate were measured with use of the headspace gas chromatography mass spectroscopy technique, as previously described by Bourdeaux et al. [46].

#### 4.4.2. Functional Lung Evaluation and Lung Edema Measurement

Dynamic lung compliance (C_dyn_) was calculated automatically by the ventilator and recorded at predefined timepoints (Appendix A). Perfusate gas was sampled hourly from the arterial and venous limb and analysis was performed with an ABL90 FLEX (Radiometer Medical ApS, Brønshøj, Denmark). Oxygenation capacity was assessed, based on the venous PO_2_-to-FiO_2_ (P/F) ratio, hourly after a 5 min evaluation phase, during which the arterial limb was deoxygenated, with use of the deoxygenation gas, which was set to 3 L/min, while the lungs were ventilated with an FiO_2_ of 100%. The difference (delta, ∆) in venous PO_2_ between the beginning and the end of EVLP was calculated as an additional indicator of oxygenation capacity.

Lung edema formation was evaluated by the total lung weight difference before EVLP (measured after 3 h SCS) vs. after EVLP, as well as the fluid content of the lung tissue samples. The fluid content was calculated and presented as a percentage of the WW (WW/DW ratio).

#### 4.4.3. Biochemical Analyses

Hyaluronan was detected by a commercial immunoassay kit (R&D Systems, Minneapolis, MN, USA), following the manufacturer’s instructions. Tumor necrosis factor-α (TNF-α) was detected using a commercial immunoassay kit (Sheep TNF-α ELISA Kit A75048, Antibodies.com, Stockholm, Sweden), also following the manufacturer’s instructions. Lactate dehydrogenase (LDH) and aspartate aminotransferase (ASAT) were tested through standard biomedical analyses in the central laboratory facility of the University Medical Center Groningen.

#### 4.4.4. RNA Isolation and Gene Expression Analysis

Tissue samples were frozen, cut and processed to prepare them for total RNA isolation with an RNeasy mini plus kit (Qiagen, Venlo, The Netherlands) according to the manufacturer’s instructions. Gel electrophoresis was used to estimate the integrity of the RNA, and yield (OD260) and purity (OD260/OD280) were tested by an Implen NanoPhotometer (Implen GmbH, München, Germany). RNA was reverse transcribed into complementary DNA using random hexamer primers (Promega, Leiden, The Netherlands) and SuperScript III (Invitrogen, Breda, The Netherlands). Assay-on-demand primer/probe sets (Taqman gene expression, Thermo Fisher Scientific, Waltham, MA, USA) (Appendix A) were used on a ViiA™ 7 Real-Time PCR System (Thermo Fisher Scientific) to perform quantitative PCR analysis (Ten ng cDNA). All the obtained duplicate cycle threshold (CT) values were averaged per sample. Hprt1 was the most stable reference gene using NormFinder algorithms in five candidate reference genes. Thus, gene expression was normalized to the expression of the reference gene Hprt1, thus calculating the ΔCT value. The average mRNA levels relative to Hprt1 were calculated by 2^−ΔCT^. For primers with the suffixes _gH, _sH and _g1, we did not detect genomic DNA in the RNA samples.

### 4.5. Statistical Analysis

Data were tested for normality with the Shapiro–Wilk test. Normally distributed, repeated measurements were analyzed with an ANOVA, and a Mann–Whitney test was used for non-normally distributed data. All data are expressed as mean ± standard deviation or as median and interquartile range. All statistical analyses were performed with RStudio (Version 1.4.1106, Posit, PBC, Boston, MA, USA) using R version 4.0.5 (The R Foundation, Vienna, Austria) and GraphPad Prism (version 9.1.0, GraphPad Software, Boston, MA, USA).

## 5. Conclusions

This is the first study showing the feasibility of ventilating lungs with sevoflurane, in a slaughterhouse EVLP model. In addition, ventilation with sevoflurane showed a potential beneficial effect on the pulmonary function during EVLP. The next step would be to study potential protective effects and points of engagement of sevoflurane in a larger and randomized setup. Our slaughterhouse EVLP model proved to be suitable to perform these studies and is a convenient setup to perform pre-clinical research on this approach.

## Figures and Tables

**Figure 1 ijms-25-02413-f001:**
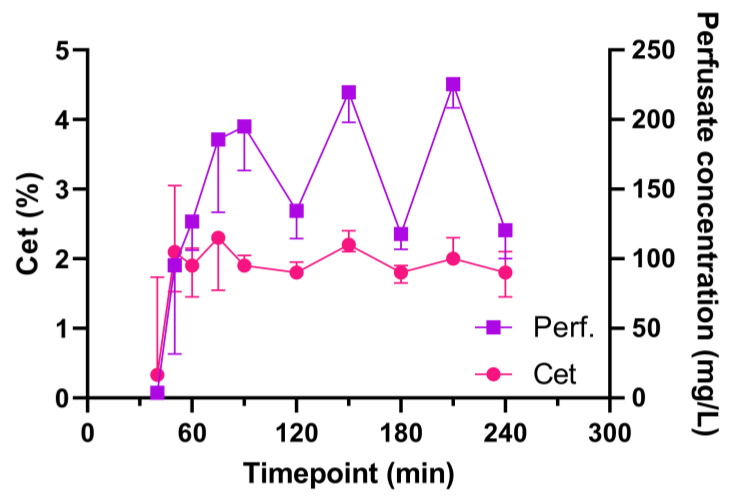
Sevoflurane end-tidal and perfusate concentrations during EVLP. Abbreviations: Cet, end-tidal concentration; Perf, sevoflurane concentration in perfusate; min, minutes after start of EVLP.

**Figure 2 ijms-25-02413-f002:**
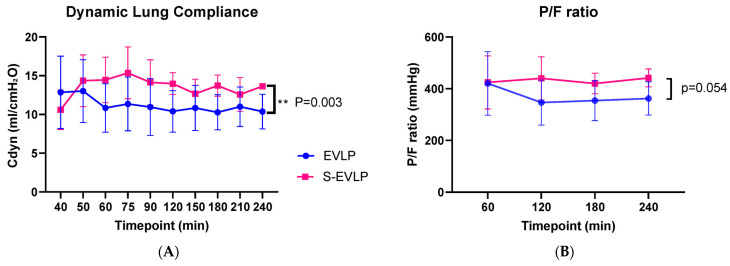
Dynamic lung compliance (**A**) and P/F ratio (**B**) during the time course of EVLP. Abbreviations: min, minutes after start of EVLP; Cdyn, dynamic lung compliance; P/F ratio, venous pO_2_/FiO_2_ ratio. ** *p* < 0.01.

**Figure 3 ijms-25-02413-f003:**
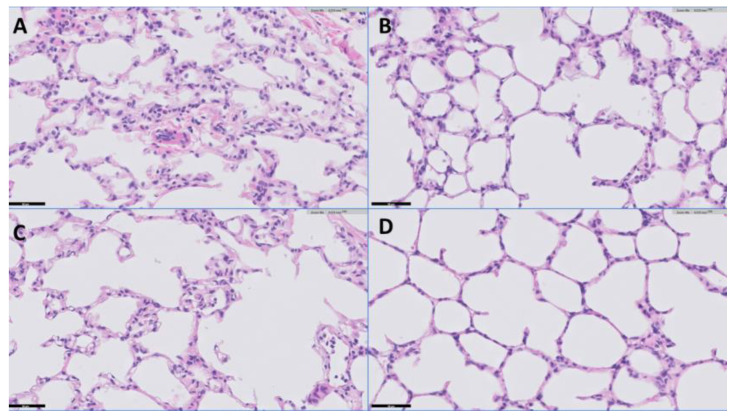
Representative example of lung histology (H&E staining). Scale indicated by the black line (=50 μm) at the left bottom of each image. (**A**): EVLP group, pre EVLP; (**B**): EVLP group, post EVLP; (**C**): S-EVLP group, pre EVLP; (**D**): S-EVLP group, post EVLP.

**Figure 4 ijms-25-02413-f004:**
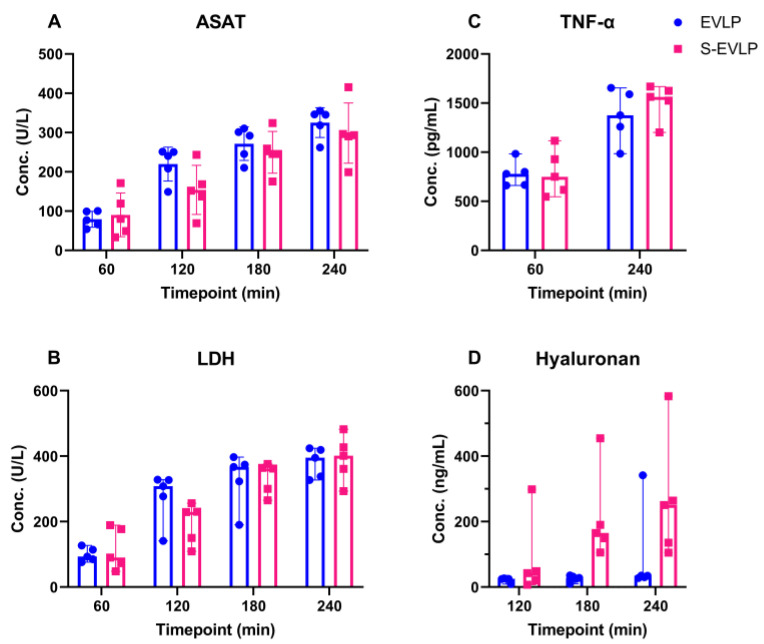
Cytokine levels in perfusate over the time course of EVLP. Abbreviations: Conc, concentration; min, minutes after start of EVLP; ASAT, aspartate aminotransferase; LDH, Lactate dehydrogenase; TNF-α, tumor necrosis factor-α.

**Figure 5 ijms-25-02413-f005:**
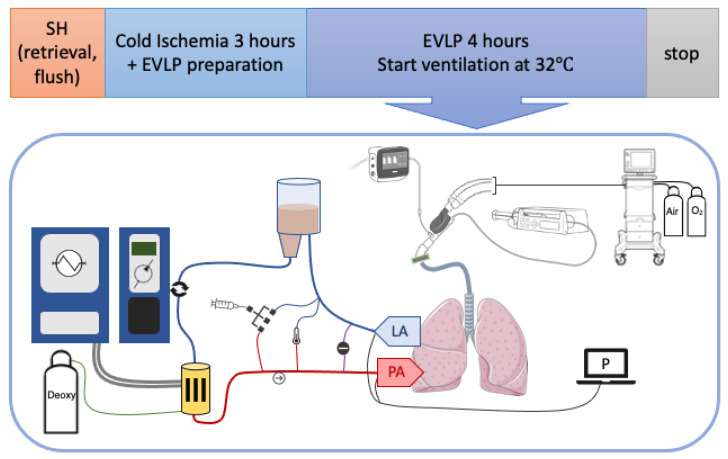
Schematic experiment flow scheme and EVLP setup. Abbreviations: SH, slaughterhouse; min, minutes after start of EVLP; LA, left atrium; PA, pulmonary artery; P, pressure measurements; Deoxy, deoxygenation gas.

**Table 1 ijms-25-02413-t001:** Baseline and EVLP characteristics.

Group	Sheep Weight (kg)	WIT (min)	SCS (min)	EVLP Duration (min)	Achieved % of Max Flow
EVLP	41.6 (41.6/46.0)	20 (18/26)	180	240 (240/250)	92.4 (81.1/96.5)
S-EVLP	46.0 (38.4/51.2)	22 (21/26)	180 (180/190)	245 (240/250)	92.9 (47.7/110.8)
*p* value	0.421	0.690	0.310	0.548	>0.99

Data are presented as median (interquartile range). Abbreviations: min, minutes; WIT, warm ischemic time (between death of the sheep and cold flush); SCS, static cold storage.

**Table 2 ijms-25-02413-t002:** Functional measurements.

Variables	S-EVLP	EVLP	*p* Value
∆pO_2_ (kPa)	3.8 (−4.9/11.1)	−11.7 (−12.0/−3.2)	0.151
P/F ratio after 4 h EVLP (mmHg)	437 (420/452)	347 (341/415)	0.056
Lung weight gain (%)	1.39 (−5.12/1.99)	6.67 (−0.22/8.78)	>0.900
∆ fluid content (%)	−6.29 (−6.35/−5.18)	1.24 (−3.31/2.43)	0.548

Data are presented as median (interquartile range). Abbreviations: pO_2_, partial oxygen pressure; P/F ratio, venous pO_2_/FiO_2_ ratio.

**Table 3 ijms-25-02413-t003:** Cytokine gene expression measurements.

	ΔPost–Pre EVLP	ΔPost–Pre S-EVLP	*p*-Value
E-selectin	0.019 (0.007, 0.067)	0.016 (0.003, 0.074)	0.691
VCAM	69.56 (29.66, 113.6)	21.33 (15.99, 199.1)	0.222
ICAM-1	0.002 (−0.094, 0.048)	−0.005 (−1.020, 0.136)	0.841
PECAM-1	6.946 ± 26.79	−9.957 ± 36.52	0.428
VE cadherin	9.354 ± 61.96	−60.92 ± 104.5	0.232
Ang1	0.254 ± 0.545	−0.931 ± 0.991	0.047
Ang2	0.531 ± 0.277	−0.251 ± 0.839	0.083
Tie2/TEK	1.404 ± 8.391	−6.371 ± 9.551	0.209
IL-6	3.385 (0.449, 6.103)	0.708 (0.394, 5.751)	0.691
TNF-α	7.71 ± 5.462	2.299 ± 2.073	0.072
MPO	0.0004 (−0.0443, 0.0479)	−0.0198 (−0.3413, −0.0028)	0.222
HIF1a	2.626 ± 3.406	−1.202 ± 9.439	0.419
VEGF-a	1.441 ± 1.469	0.983 ± 3.035	0.769
E-selectin	0.019 (0.007, 0.067)	0.016 (0.003, 0.074)	0.691
VCAM	69.56 (29.66, 113.6)	21.33 (15.99, 199.1)	0.222

Pre tissue samples were obtained from the cranial part of the right upper lobe; post tissue samples were obtained from the caudal part of the left upper lobe (Appendix A [30]). Data are presented as median (interquartile range) or mean ± standard deviation, depending on their respective distribution. Abbreviations: Δ, delta, difference; VCAM, vascular cell adhesion molecule; ICAM-1, intercellular adhesion molecule 1; PECAM-1, platelet and endothelial cell adhesion molecule 1; VE cadherin, vascular endothelial cadherin; Ang1, angiopoietin 1; Ang2, angiopoietin 2; Tie2/TEK, TEK receptor tyrosine kinase; IL-6, interleukin 6; TNF-α, tumor necrosis factor alpha; MPO, myeloperoxidase; HIF1a, hypoxia-inducible factor 1 subunit alpha; VEGF-a, vascular endothelial growth factor A.

## Data Availability

The data that support the findings of this study are available from the corresponding author upon reasonable request.

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
