# Peer review of "Ex Vivo Optimization of Donor Lungs with Inhaled Sevoflurane during Normothermic Ex Vivo Lung Perfusion (VITALISE): A Pilot and Feasibility Study in Sheep"

_ijms, 2024, doi:10.3390/ijms25042413_

Round 1
Reviewer 1 Report
Comments and Suggestions for Authors
This manuscript demonstrates the role of sevoflurane in improving lung quality during ex vivo perfusion by preclinical studies. Here are some questions that the authors should comprehensively address to support their findings.
1. Please clarify why you choose sevoflurane as one of the representatives of the volatile anaesthetics for graft reperfusion. What's the rationale? Does other anaesthetics work?
2. In Figure 1, the authors should indicate the reasons why the perfusion concentration of sevoflurane varied (around 2.5% at 60, 120, 180, and 240 mins and around 4% at 90, 150, and 210 mins).
3. In the study design, lung function was evaluated after perfusion but not after transplant. It is suggested to detect the lung function after transplant. The levels of ASAT, LDH and TNG-alpha in the blood can be compared between EVLP and S-EVLP groups.
4. As shown in the supplementary Figure 2, the lung samples were collected from different lobes. The authors should indicate the cytokine gene expression in different lobes in Table 3.
5. Why do you define a target concentration of sevoflurane as 2%? The authors should perform quality control.
Author Response
Dear reviewer,
We would like to thank you for your valuable comments and feedback. Hereby we answer to your remarks and questions point by point.
- Please clarify why you choose sevoflurane as one of the representatives of the volatile anaesthetics for graft reperfusion. What's the rationale? Does other anaesthetics work?
Very valid point. We’ve now addressed this in the discussion:
Since this first step was a feasibility study, we chose to perform 4 hours of perfusion, with a sevoflurane Cet of 2% although it has been shown that EVLP can be performed for up to 12 hours [4]. Accordingly, future studies could consider studying longer perfusion times, potentially enhancing the beneficial effects of sevoflurane. In addition, higher concentrations of sevoflurane should be investigated to study dose dependent effects. In step 2 of the VITALISE project we’re currently comparing 0%, 2%, 4% and 6% Cet of sevoflurane in a randomized setting. Since sevoflurane has shown superior effects over isoflurane in pulmonary inflammation models and liver transplant models, we choose to use sevoflurane in this project [29, 41]. Since desflurane showed the least organ protective effects in various experiments and its use in clinical practice is phased out due to its significant global warming potential we don’t find it justified to study it’s effect during EVLP [42]. Isoflurane could be taken into account in future experiments and compared to sevoflurane in which the focus should lie on differences in their protective potential and specific mechanisms of action.
- In Figure 1, the authors should indicate the reasons why the perfusion concentration of sevoflurane varied (around 2.5% at 60, 120, 180, and 240 mins and around 4% at 90, 150, and 210 mins).
This was indeed a very interesting outcome that we also did not anticipate. We’ve addressed this finding in the discussion.
Sevoflurane levels in the perfusate markedly decreased at 60, 120, 180 and 240 min, which was also reflected in lower Cet values (Figure 1). At these timepoints the oxygenation capacity of the lung was evaluated, and the perfusion solution was actively deoxygenated, whilst the FiO2 was set to 100%. This might have caused a second gas effect, leading to an increased outwards diffusion of sevoflurane along with the outward diffusion of O2 in the membrane (de)oxygenator [30].
In our step 2 of the project, we’ve adapted the sampling timepoints. We’ve clarified the figure by explaining the abbreviation ‘Perf.’
- In the study design, lung function was evaluated after perfusion but not after transplant. It is suggested to detect the lung function after transplant. The levels of ASAT, LDH and TNG-alpha in the blood can be compared between EVLP and S-EVLP groups.
We agree that ideally, inflammatory markers are also measured after transplantation in blood. However, our model uses slaughterhouse organs which cannot be transplanted into a living animal after perfusion. If sevoflurane proves its protective effects against IRI we would like to translate this experimental model to a more advanced one, which will also allow transplantation after EVLP.
- As shown in the supplementary Figure 2, the lung samples were collected from different lobes. The authors should indicate the cytokine gene expression in different lobes in Table 3.
This is indeed an important sidenote when interpreting the histological findings. As the tissue samples obtained before EVLP are from a different location of the lung than the post-EVLP biopsies, this also makes comparison difficult. The fact, that we observed no notable (deterioration of) lung injury however supports the interpretation that our EVLP model does not cause or worsens significant short-term histological injury to the lungs. We’ve added origin of the samples in the footnote of table 3.
- Why do you define a target concentration of sevoflurane as 2%? The authors should perform quality control.
This question is now as well addressed in the discussion. For this pilot study we’ve decided to start the project with a concentration used in clinical settings, of which we know is safe, in order to get an idea about the feasibility and challenges. The study of Wang et al (Reference 21) used a sevoflurane concentration of 2% and we as well anticipated a Cet of 2% to be high enough to give an indication about possible effects of sevoflurane. In step 2 of the VITALISE project we’re currently comparing 0%, 2%, 4% and 6% Cet of sevoflurane in a randomized setting to see whether there are dose dependent differences in the protective potential.
Reviewer 2 Report
Comments and Suggestions for Authors
The authors report on the feasibility of the application of volatile anesthetics (Sevoflurane) in an ex vivo perfusion model of sheep lungs. Even if the treated group showed significantly better dynamic lung compliance compared to untreated controls, no significant impact on inflammation could be observed.
Machine perfusion of donor organs is an increasing field, hence opening up the possibility of organ optimization and reconditioning. Decreasing IRI and inflammatory mediators, which may trigger further inflammation during ex vivo organ perfusion, is a major goal.
The manuscript is well-written.
A relatively low N is a major hurdle in reaching statistical significance in numerous studies conducted in this field.
I was wondering why the perfusion end point was set to 4h, while it is reported that lungs can be perfused relatively stable for up to 12h. As the authors stated, this short perfusion time could have been the reason for not having detected tissue damage in histology samples. What were the reasons for having chosen 4h of perfusion?
Even if damage was not obvious in tissue biopsies at 4h, the authors could check for the expression of damage markers on mRNA level (eg Caspase 3....) to assess for differneces beween groups.
Did the authors assess for ROS levels or surrogate markers involved in this pathway? This may also help to detect for an early impact of Sevoflurance on IRI/inflammation in tissue samples.
As the authors have sampled perfusate at very close intervals it may be advisable to assess perfusate cytokines and DAMPs in more detail.
Do the authors think that using a higher concentration of Sevoflurance could have a more beneficial impact on inflammation/cytokines?
Major: The authors stated that at the end of the study (when they did the controls) the team was more experienced and trained, compared to the beginning of the study, when they did the S-EVLP group. I would therefore recommend to do another 2-3 perfusions of this group in order to balance this limitation/drawback of the study.
Minor: Table 3: S-EVLP group?
Author Response
Dear reviewer,
We would like to thank you for your valuable comments and feedback. Hereby we answer to your remarks and questions point by point.
- A relatively low N is a major hurdle in reaching statistical significance in numerous studies conducted in this field.
We completely agree with the reviewer. Since this first step was designed as a feasibility trial to test whether ventilation with sevoflurane during EVLP is feasible, and to test whether sevoflurane is taken up by the perfusion solution, we didn’t aim to detect differences between degree of injury and inflammation. In the currently performed second step of the project, comparing different concentrations of sevoflurane, we’ve expanded the group size and randomized lungs.
- I was wondering why the perfusion end point was set to 4h, while it is reported that lungs can be perfused relatively stable for up to 12h. As the authors stated, this short perfusion time could have been the reason for not having detected tissue damage in histology samples. What were the reasons for having chosen 4h of perfusion?
Very valid point. We’ve now addressed this in the discussion section:
Since this first step was a feasibility study, we chose to perform 4 hours of perfusion, with a sevoflurane Cet of 2% although it has been shown that EVLP can be performed for up to 12 hours [4]. Accordingly, future studies could consider studying longer perfusion times, potentially enhancing the beneficial effects of sevoflurane.
For this study, we had a rather small executing team, seen that the experiments require constant attention from three to five people. This team had to perform all experiments including the procurement in the slaughterhouse until the perfusion was finished and everything was cleaned up. To do longer perfusions, a significantly larger team is required. For the future studies, that is indeed planned, and we are currently busy, training more people to run the experiments. Investigating the differences between various perfusion durations is one of the next steps in this project.
- Even if damage was not obvious in tissue biopsies at 4h, the authors could check for the expression of damage markers on mRNA level (eg Caspase 3....) to assess for differneces beween groups.
- Did the authors assess for ROS levels or surrogate markers involved in this pathway? This may also help to detect for an early impact of Sevoflurance on IRI/inflammation in tissue samples.
- As the authors have sampled perfusate at very close intervals it may be advisable to assess perfusate cytokines and DAMPs in more detail.
For this pilot study, we indeed did not investigate every possible marker in much detail. We tried to focus on the most important analyses to get an idea about the capabilities of our model. We’ve performed mRNA analysis of various injury and inflammation related genes in our tissue samples (supplemental material). In addition, we’ve tried to stain our tissue samples for caspase 3 which was difficult due to cross reaction of the antibody with other proteins and limited availability of antibodies for sheep material. In step 2 of our project, we will look into more detail by using precision cut slice models after EVLP. Also, during the EVLPs we will assess cytokines, DAMPs and ROS in more detail. Thereby, we hope to get more insight in the involved pathways. However, this extends the scope of this feasibility project.
Regarding ROS this is off course difficult to assess. Since a substantial part will be intracellular. In step 2 we’ll study mitochondrial viability and ROS target products like MDA-TBAR. Since the assessment of degree of inflammation was not the primary aim of this feasibility study, we’ve limited the amount of sample timepoints for analysis due to high costs of Elisa’s.
- Do the authors think that using a higher concentration of Sevoflurance could have a more beneficial impact on inflammation/cytokines?
The concentration is another interesting point. A paragraph on this is now added to the discussion:
In addition, higher concentrations of sevoflurane should be investigated to study dose dependent effects. In step 2 of the VITALISE project we’re currently comparing 0%, 2%, 4% and 6% Cet of sevoflurane in a randomized setting.
We decided for this pilot study to start the project with rather short perfusion duration and low concentration in order to get an idea about the feasibility and challenges. The study of Wang et al [21] used a sevoflurane concentration of 2% and we as well anticipated a Cet of 2% to be high enough to give an indication about possible effects of sevoflurane. In step 2 of the project, we focus on difference between various concentrations to find optimum dosage for the research steps thereafter.
- Major: The authors stated that at the end of the study (when they did the controls) the team was more experienced and trained, compared to the beginning of the study, when they did the S-EVLP group. I would therefore recommend to do another 2-3 perfusions of this group in order to balance this limitation/drawback of the study.
Lastly, you suggested to perform 2-3 more S-EVLP experiments to compensate for the learning curve. This is a fair point which we initially also considered. However, as we observed such a big variation depending on the season and the supplying farms, we anticipated very different results from those added experiments. The experiments need to take place in the same season in order to get somewhat comparable results. As we were still able to prove the feasibility and learned what needed to be adapted and replanned for step 2 of the project, looking into protective properties of different dosages and duration, we decided not to do another 2-3 perfusions. Randomization in step 2 will in turn also compensate for the seasonal effects. For the aim of step 1 of the project, namely, to investigate the feasibility of the methodology, the addition of more experiments seemed not strictly necessary.
- Minor: Table 3: S-EVLP group?
Thanks for pointing this out. We’ve adapted this.
Reviewer 3 Report
Comments and Suggestions for Authors Although the mentioned study is a pilot study, the sample size is low. Comments on the Quality of English Language Minor corrections are required in terms of English grammarAuthor Response
Dear reviewer,
Thank you for your valuable feedback. We completely agree with the reviewer. Since this first step was designed as a feasibility trial to test whether ventilation with sevoflurane during EVLP is feasible and to test whether sevoflurane is taken up by the perfusion solution we didn’t aim to detect differences between degree of injury and inflammation. In the currently performed second step of the project comparing different concentrations of sevoflurane we’ve expanded the group size and randomize lungs.
We corrected some last grammar flaws that we spotted thanks to your feedback.
Round 2
Reviewer 1 Report
Comments and Suggestions for Authors
Good.
Reviewer 2 Report
Comments and Suggestions for Authors
I would like to thank the Authors for their valuable feedback and their modifications. I agree with their suggestions and would like to wish them all the best for the second phase of the study. I do not have any further comments.